METHODS AND RESOURCES

# FURNA: A database for functional annotations of RNA structures

Chengxin Zhang [1,2]*, Lydia Freddolino [1,2]*

**1** Department of Computational Medicine and Bioinformatics, University of Michigan, Ann Arbor, Michigan, United States of America, **2** Department of Biological Chemistry, University of Michigan, Ann Arbor, Michigan, United States of America

* zcx@umich.edu (CZ); lydsf@umich.edu (LF)

## Abstract

Despite the increasing number of 3D RNA structures in the Protein Data Bank, the majority of experimental RNA structures lack thorough functional annotations. As the significance of the functional roles played by noncoding RNAs becomes increasingly apparent, comprehensive annotation of RNA function is becoming a pressing concern. In response to this need, we have developed FURNA (Functions of RNAs), the first database for experimental RNA structures that aims to provide a comprehensive repository of high-quality functional annotations. These include Gene Ontology terms, Enzyme Commission numbers, ligand-binding sites, RNA families, protein-binding motifs, and cross-references to related databases. FURNA is available at https://seq2fun.dcmb.med.umich.edu/furna/ to enable quick discovery of RNA functions from their structures and sequences.

## Introduction

Advances in experimental RNA structure determination methods, particularly Cryo-EM [1], have resulted in over 16,000 RNA chains being deposited into the Protein Data Bank (PDB) database [2]. Despite these strides, functional annotations of experimental RNA structures are glaringly absent in both the PDB and secondary databases. The PDB database merely includes the bare minimum annotations for RNA chains, such as their names, lengths, and species. Downstream databases like NAKB (formerly known as NDB) [3], DNATCO [4], and BGSU RNA [5] offer more annotations for base pairing, backbone torsions, and 3D motifs, yet the annotation of the RNAs' biological roles is still wanting. The MeRNA database [6] was probably the only database dedicated to function (in this case, metal ion binding sites) for experimental RNA structures, but it has long been defunct and was limited to 256 RNAs and their binding to metal ions. Simultaneously, recent studies have confirmed that many noncoding RNAs play vital roles in numerous biological events, particularly those involved in gene expression regulations [7], making RNA structures ideal targets for drug design [8]. This fresh understanding underscores the importance of annotating RNA functions for the RNA biology community.

In contrast to the stark lack of a functional database for RNA structures, several databases to annotate protein functions have already been established. Databases such as PDBsum [9]

**Data Availability Statement:** All necessary data are provided for download in the FURNA database itself, which is the subject of the manuscript. Source code for establishing the FURNA database is available for download from https://github.com/

freddolino-lab/furna/. The database itself is available for download from https://doi.org/10.5281/zenodo.11664059 and https://doi.org/10.5281/zenodo.11672037 (split into two files due to size limitations).

**Funding:** This work was directly supported by NIH R01 AI134678 (to LF). This work also used the Advanced Cyberinfrastructure Coordination Ecosystem: Services & Support (ACCESS) program, which is supported by National Science Foundation (2138259, 2138286, 2138307, 2137603, and 2138296). The funders played no role in the study design, data collection, manuscript preparation, or decision to publish.

**Competing interests:** LF is a paid consultant and scientific advisory board member for Circnova, Inc. Circnova did not support the work in any way, and played no role in the study design, performance, or decision to publish.

**Abbreviations:** BP, Biological Process; CC, Cellular Component; CCD, Chemical Component Dictionary; EC, Enzyme Commission; GO, Gene Ontology; HDV, hepatitis delta virus; IL, incremental length; MAD, multiwavelength anomalous diffraction; MF, Molecular Function; PDB, Protein Data Bank; PWM, position weight matrix; TPP, thiamine pyrophosphate.

and SIFTS [10] annotate protein chains in the PDB using Gene Ontology (GO) terms and Enzyme Commission (EC) numbers by mapping PDB chains to UniProt [11] proteins and InterPro [12] families. The PDBbind-CN [13], BindingDB [14], and Binding MOAD [15] databases collect protein–ligand interactions with known affinity data. The PDBe-KB [16] database features ligand-binding sites and posttranslational modification sites for all PDB proteins. The FireDB [17] and IBIS [18] database curate protein–ligand interaction data in the PDB. Most recently, BioLiP2 [19] was developed as a comprehensive database covering almost all functional aspects of PDB proteins, including GO terms, EC numbers, ligand-binding sites, binding affinities, and cross-reference to external databases.

Inspired by BioLiP2, we created FURNA, the first database in the field to offer comprehensive functional annotations for all RNA chains in the PDB database. Function annotations in FURNA include GO terms, EC numbers, Rfam [20] RNA families, RNA motifs for protein binding, species, literature, and cross references to external databases like PDBsum, NAKB, DNATCO, BGSU RNA, ChEMBL [21], DrugBank [22], ZINC [23], and RNAcentral [24]. Unlike protein–ligand interaction databases such as BioLiP, FireDB, and PDBbind-CN, which consider receptor–ligand contacts within each asymmetric unit, FURNA determines RNA–ligand interactions based on the biological assembly (i.e., biounit). This approach situates RNA–ligand interactions within the context of its quaternary structure (i.e., the RNA's interaction with nucleic acid and protein partners). FURNA is available both as an open-source software package and as a browsable and searchable web service at https://seq2fun.dcmb.med.umich.edu/furna/.

## Results

### Overall statistics

At the time of writing this manuscript (October 2023), FURNA includes 16154 RNAs involved in 380680 ligand–RNA interactions; the online version of the database is updated on a weekly basis. Among these interactions, 186025, 138245, 31659, 24056, and 695 are interactions with metal ions, proteins, "regular" small molecule compounds excluding metal ions, other RNAs, and DNAs, respectively. Unlike BioLiP, FURNA does not attempt to exclude "biologically irrelevant" RNA-associated molecules from the database apart from removal of water molecules. This is because the biological relevance of ligands, especially metal ions, are less clearly defined for RNAs than for proteins. For example, calcium ions ($Ca^{2+}$) are usually biologically irrelevant artifacts added for purification and/or crystallization purposes for proteins, but they are used to substitute magnesium ions ($Mg^{2+}$) that are critical to maintain the folding of RNAs in pre-catalytic states [25]. Similarly, while potassium ions ($K^+$) are a simple buffer additive for many proteins, they are critical for the folding of many large RNAs where potassium ions stabilize juxtaposition of nucleotides with large sequence separation by neutralizing charge density [26]. This is why a significant portion (48.9%) of ligand–RNA interactions in FURNA are metal ions, among which 91.4% are magnesium ions, which are the most critical ion for RNA folding (**Fig 1A**). For small molecule ligands that are not metal ions, the 2 most frequent compounds are osmium (III) hexammine and cobalt hexammine (III) (**S1 Table**), which are crystallization additives used to determine the RNA structure by multiwavelength anomalous diffraction (MAD) phasing [27]. Although these compounds are metal-containing coordination complexes with positive charges, FURNA does not consider them to be metal ions because they are not natural ions, do not bind to natural binding sites of biologically relevant metal ions (**S1 Fig**), and do not consist solely of a single metal atom.

Among the 10561 RNAs with GO annotations in FURNA, 9288, 5311, and 7014 have annotations in Molecular Function (MF), Biological Process (BP), and Cellular Component (CC)

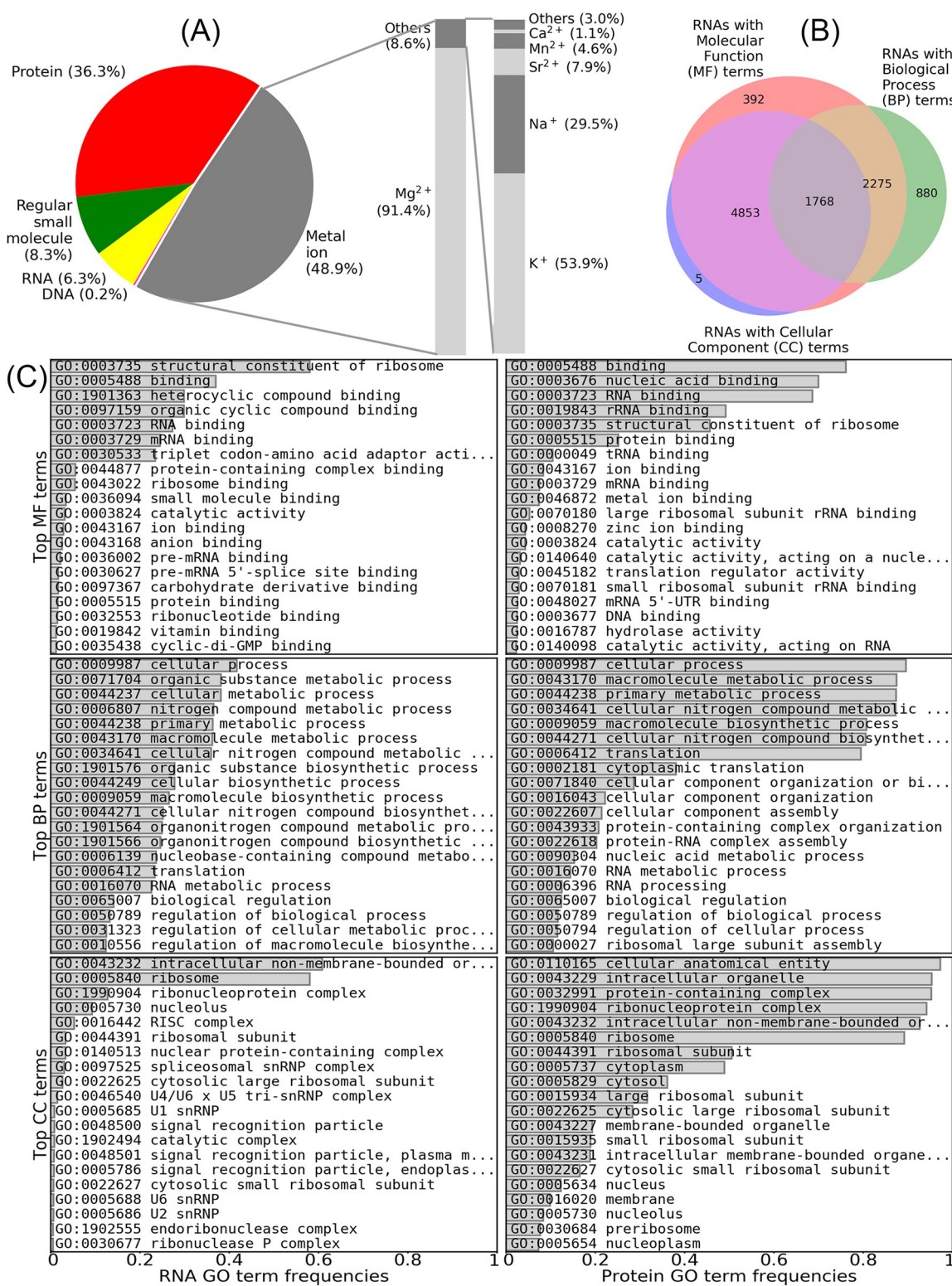

**Fig 1. Overall statistics for RNAs and ligand–RNA interactions in FURNA.** (A) Pie chart for the breakdown of ligand types in ligand–RNA interactions. (B) Venn graph showing the numbers of RNAs with GO terms in the MF, BP, and/or CC aspects. (C) Top GO terms in RNAs and proteins collected by FURNA. Data for panels (A) and (B) are available under the section "Database statistics" from the home page of the FURNA website.

aspects, respectively (**Fig 1B**). Out of the RNAs with MF terms, 58.0% are rRNAs (denoted by GO:0003735 "structural constituent of ribosome") and 23.5% are tRNAs (indicated by GO:0030533 "triplet codon-amino acid adaptor activity") (**Fig 1C**). This suggests that the distribution of RNA families among experimentally determined RNA structures is highly biased, consistent with MF annotations for RNA-binding proteins where GO:0019843 "rRNA binding" and GO:0000049 "tRNA binding" are among the most common GO terms. It is worth noting that, on average, the similarity of BP GO term annotations between an interacting RNA–protein pair is significantly higher than a random RNA-RNA pair or a random RNA–protein pair (**S2 Fig**, Mann–Whitney U test $p$-value <1E-300 and $p$-value = 1.0E-20, respectively). This suggests that RNA–protein interactions will be useful for RNA BP term prediction, similar to the utility of protein–protein interactions in protein function prediction [28–30].

## Web interface

The FURNA website provides 3 primary interfaces: SEARCH, BROWSE, and DOWNLOAD. The functionalities of these interfaces are elaborated upon below.

## BROWSE

Each entry in the FURNA database represents 1 RNA chain in the PDB. For each of these RNA chains, the BROWSE interface displays the PDB ID and chain ID, resolution, EC number, GO terms, RNAcentral accessions, Rfam families, species, PubMed citations, and protein-binding motifs found in the ATtrRACT [31] database (**Fig 2**). Additionally, if the RNA chain has a non-water ligand, the BROWSE interface also provides information on the ligand ID, the chain and residue sequence number of the ligand, the ligand-binding nucleotides on the query RNA, and the biological assembly information where interaction with that ligand was retrieved.

Individual pages, accessed by clicking on the ligand in the last column of the BROWSE table, offer detailed structure and function information of each ligand–RNA interaction. These individual pages include the 3D structures of the RNA chain on its own, the full biological assembly, the RNA–ligand pair, and the local structure of the ligand binding site. These are displayed via 4 JSmol [32] applets, where the second applet uses the mmCIF-formatted structure while the remaining 3 applets use PDB format structures. The reason for this difference is that the full biological assembly may contain more than 99,999 atoms and/or 62 chains, making it impossible to be represented in PDB format [33]. On the other hand, an individual RNA–ligand pair can always be represented by a PDB file, which has a smaller file size than an mmCIF file [34] and is compatible with more bioinformatics tools [33]. Where available, GO terms for Molecular Function, Biological Process, and Cellular Component are presented in 3 directed acyclic graphs created using Graphviz [35], illustrating the relationships among different GO terms. The GO terms, as well as EC numbers, are also listed as tables. Additional information is also provided, including the RNA sequence and secondary structure, resolution, the structure's name, species, ATtRACT motifs, PubMed citations, and crosslinks to other databases. In case of a small molecule ligand or an ion, the page exhibits the 2D diagram, ligand IDs (including PDB CCD ID, ChEMBL ID, DrugBank ID, and ZINC ID), the chemical formula, ligand name, and linear descriptions of the molecules (**Fig 3**). For RNA, DNA, or protein ligands, additional details such as the sequence, name, and species, as well as relevant function annotations such as GO terms and EC numbers, are provided when available (**S3 Fig**). In addition to webpages for individual ligand–RNA interactions, FURNA also provides individual webpages for each RNA chain (**S4 Fig**), which can be viewed by clicking on the first column of the summary table at the BROWSE interface (**Fig 2**). In addition to showing the 3D

| # | PDB (Resolution) | Length | EC number & GO term | RNAcentral | Rfam | Taxon | PubMed | ATtRACT motif | Ligand & binding nucleotides |
|---|---|---|---|---|---|---|---|---|---|
| 1 | 1na2:A | 30 | EC:2.7.7.49 RNA-directed DNA polymerase<br>GO:0003720 (F) telomerase activity<br>GO:0007004 (P) telomere maintenance vi...<br>GO:0000333 (C) telomerase catalytic co... | URS000080DF2DRF00024 | | Homo sapiens 9606 | 12525685 | s65<br>58<br>s66<br>122<br>1188<br>1125 | |
| 2 | 1nbs:A (3.15Å) | 120 | EC:3.1.26.5 ribonuclease P<br>GO:0004526 (F) ribonuclease P activity<br>GO:0008033 (P) tRNA processing<br>GO:0030680 (C) dimeric ribonuclease P ... | URS000080E089 | RF00011<br>RF01100 | Bacillus subtilis 1423 | 12610630 | | Assembly1<br>PB:A:241 186,219~220<br>PB:A:242 90,132<br>PB:A:244 134,181~182<br>PB:A:249 228~229 |

**Fig 2. Summary table for browsing function annotations in FURNA.**

structure of the RNA chain on its own and the full biological assembly, it also lists the sequence, secondary structure, resolution, name, species, ATtRACT motifs, PubMed citations, crosslinks to other databases and, whenever available, associated GO terms and EC numbers. If the RNA is a ribozyme with known catalytic active site(s), the active site nucleotides are listed as a table and highlighted in the structure applet (**S4 Fig**). All ligands for the subject RNA chain are tabulated under the "Interaction partners" section to provide links to view the individual ligand–RNA interaction pages (**Fig 3**).

## SEARCH

The "SEARCH" interface provides 4 methods to explore FURNA: "Search by name," "Quick sequence search (via BLAST)," "Sensitive sequence search (via Infernal)," and "Search by structure." Firstly, users can query FURNA using PDB ID, PDB chain ID, ligand ID (as defined by the 3-letter code in the PDB database's Chemical Compound Dictionary), ligand name, RNAcentral accession, Rfam family, EC number, GO term, ATtRACT motif, taxonomy, PubMed ID, or any combination of these. Secondly, FURNA can employ NCBI BLAST+ to search its entries using RNA, DNA, or protein sequences through a local nonredundant sequence database where identical sequences are merged into the same entry. In the search results, both representative hits found in the nonredundant database and members from the same sequence clusters are displayed (**Fig 4A**). Thirdly, to address the issue of a BLAST search's low sensitivity for nucleic acid sequences, FURNA offers an alternative, more sensitive RNA sequence search option using Infernal (see Materials and methods, **Fig 4B and 4C**). Lastly, users can search the tertiary structure of a query RNA (in PDB format) through the FURNA database using US-align (see Materials and methods).

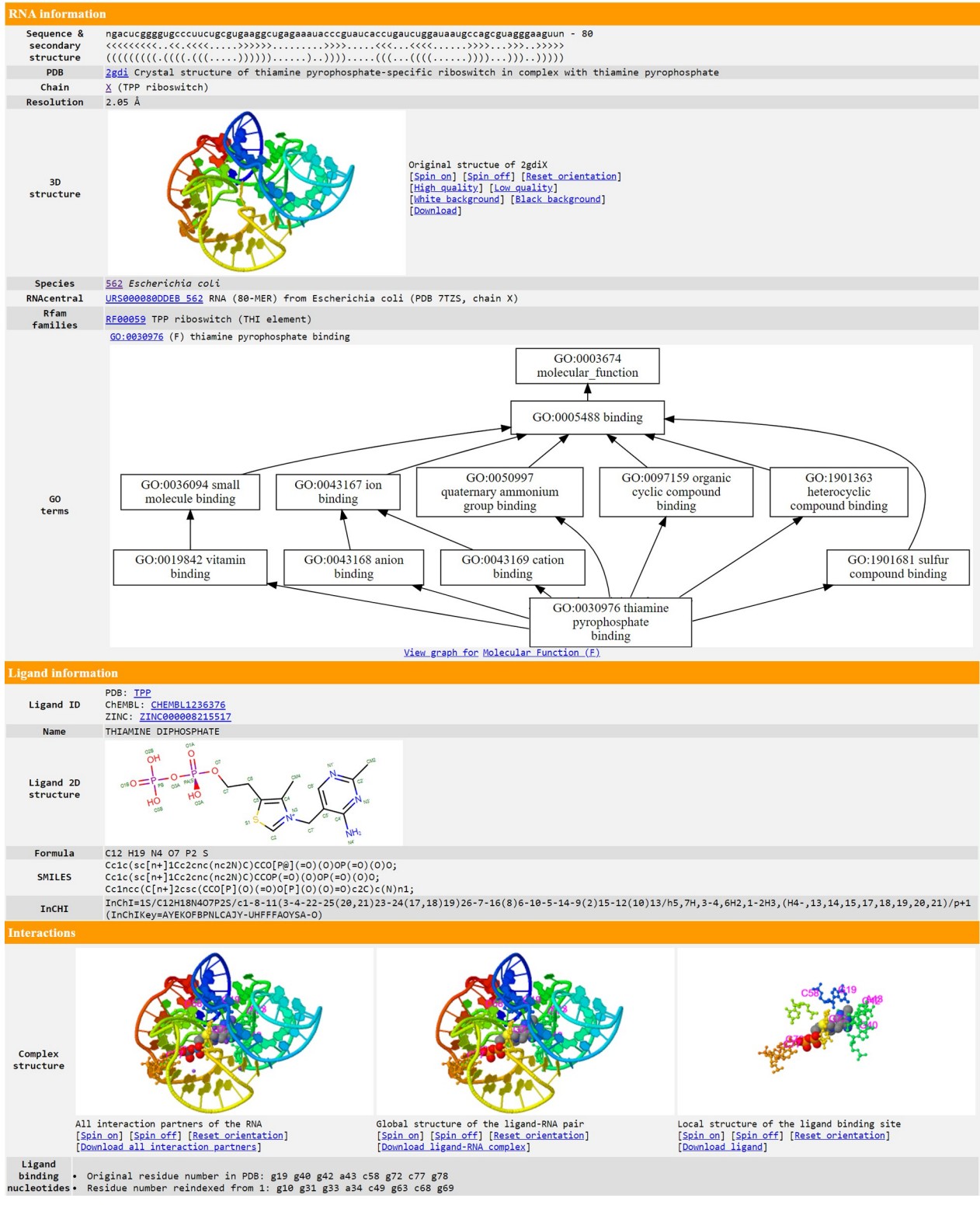

**Fig 3. Excerpt of an individual FURNA web page for a small molecule-RNA interaction.** Shown is the FURNA web page for the interaction between thiamine diphosphate and TPP riboswitch (PDB 2gdi chain X, https://seq2fun.dcmb.med.umich.edu/furna/pdb.cgi?pdbid=2gdi&chain= X&lig3=TPP&ligCha=X&ligIdx=1). The structures of the RNA with all interaction partners, the RNA with only the ligand of interest, and the ligand itself without the RNA can be downloaded through the links at the bottom of the "Interaction" section.

(A)

| # | Hit | Hit length | Aligned length | Identity (normalized by query) | Identity (normalized by hit) | Identity (normalized by aligned length) | E-value | Homologs to hit |
|---|---|---|---|---|---|---|---|---|
| 1 | 2hoo:A Crystal structure of an E. coli thi-box riboswitch bound to benfotiamine; thi-box riboswitch | 83 | 79 | 0.3950 | 0.9518 | 1.0000 | 2.98e-36 | |
| 2 | 2gdi:X Crystal structure of thiamine pyrophosphate-specific riboswitch in complex with thiamine pyrophosphate; TPP riboswitch | 80 | 77 | 0.3850 | 0.9625 | 1.0000 | 3.85e-35 | 2gdi:Y, 7tzs:X, 7tzs:Y, 4nya:A, 4nya:B, 7tzr:X, 7tzr:Y |
| 3 | 2hom:A Crystal structure of an E. coli thi-box riboswitch bound to thiamine monophosphate; thi-box riboswitch | 80 | 79 | 0.3800 | 0.9500 | 0.9620 | 3.88e-30 | |
| 4 | 7tda:A Crystal structure of the E. coli thiM riboswitch in complex with thiamine pyrophosphate, manganese ions; thiM TPP riboswitch RNA (80-MER) | 80 | 79 | 0.3800 | 0.9500 | 0.9620 | 3.88e-30 | 7tdc:A, 7tdb:A, 4nyd:A, 4nyb:A, 4nyc:A |
| 5 | 7tzu:A Crystal structure of the E. coli thiM riboswitch bound to 1-(4-(piperazin-1-yl)pyridin-3-yl)-N-(quinoxalin-6-ylmethyl)methanamine (linked compound 38); RNA (79-MER) | 79 | 79 | 0.3750 | 0.9494 | 0.9494 | 1.81e-28 | 2hol:A, 7tzt:A |

(B)

| # | Hit | Length | Strand | E-value | Homologs to hit |
|---|---|---|---|---|---|
| 1 | 2hoo:A Crystal structure of an E. coli thi-box riboswitch bound to benfotiamine; thi-box riboswitch | 83 | + | 1.8e-26 | |
| 2 | 2gdi:X Crystal structure of thiamine pyrophosphate-specific riboswitch in complex with thiamine pyrophosphate; TPP riboswitch | 80 | + | 3.8e-26 | 2gdi:Y, 7tzs:X, 7tzs:Y, 4nya:A, 4nya:B, 7tzr:X, 7tzr:Y |
| 3 | 7tda:A Crystal structure of the E. coli thiM riboswitch in complex with thiamine pyrophosphate, manganese ions; thiM TPP riboswitch RNA (80-MER) | 80 | + | 3.7e-23 | 7tdc:A, 7tdb:A, 4nyd:A, 4nyb:A, 4nyc:A |
| 4 | 2hoj:A Crystal structure of an E. coli thi-box riboswitch bound to thiamine pyrophosphate, manganese ions; thi-box riboswitch | 79 | + | 1.5e-22 | 4nyg:A |
| 5 | 7tzu:A Crystal structure of the E. coli thiM riboswitch bound to 1-(4-(piperazin-1-yl)pyridin-3-yl)-N-(quinoxalin-6-ylmethyl)methanamine (linked compound 38); RNA (79-MER) | 79 | + | 7.1e-22 | 2hol:A, 7tzt:A |

**Fig 4. FURNA database sequence search results for the *E. coli thiM* riboswitch.** (A) Top 5 BLAST search hits. (B) Top 5 Infernal search hits.

## DOWNLOAD

All data from FURNA can be downloaded in bulk through the "DOWNLOAD" interface. Functional annotations for each RNA chain and each ligand–RNA interaction are available in tab-separated tables. The FASTA sequences of RNAs, plus those of RNA-binding proteins and RNA-binding DNAs, are also provided. The coordinates of the RNAs and all non-water ligands are supplied in PDB format files. Furthermore, the link to the source codes for database curation and website display is also located on this page.

## Case study on TPP riboswitches

To illustrate FURNA's utility in RNA function annotation, we conducted a case study involving the TPP (thiamin pyrophosphate)-binding riboswitches, also known as the THI element or

Thi-box riboswitch. This well-known family of riboswitches binds to thiamine pyrophosphate (TPP) to regulate the expression of its downstream gene [36,37]. In *Escherichia coli*, one such riboswitch is located upstream of the Hydroxyethylthiazole kinase (*thiM*) coding sequence [38,39] (**Fig 3** and **S2 Table**). We used the 200 nucleotides immediately upstream of the *thiM* open reading frame as the query sequence to search FURNA. Unsurprisingly, both BLAST (**Fig 4A**) and Infernal (**Fig 4B**) searches of the *E. coli* TPP riboswitch through FURNA return hits for many TPP riboswitches, including those from *E. coli*. Similar results can be obtained by searching the region upstream of the *thiM* gene of *Siccibacter turicensis*, which also belongs to the Enterobacteriaceae family (**S2 Table**).

Based on gene function and the general prevalence of Thi-box riboswitches, we suspected the presence of riboswitches at several locations in *Bacillus subtilis*, e.g., one situated upstream of the coding sequence of the HMP/thiamine-binding protein (*ykoF*) and the other situated upstream of the aminopyrimidine aminohydrolase (*tenA*, **S2 Table**). Indeed, the *tenA* riboswitch has been previously reported [40], whereas a riboswitch upstream of *ykoF* has not, to our knowledge, been previously reported in the literature, although its presence is indicated in the RNAcentral (https://rnacentral.org/rna/URS000005CA97) and Rfam (https://rfam.org/family/RF00059) databases. When using FURNA to perform a BLAST sequence search of the putative *B. subtilis* TPP riboswitches (i.e., the 200 nucleotides immediately upstream of the *ykoF* and *tenA* open reading frames), no hits are returned, including hits to the *E. coli thiM* riboswitch. This outcome is not unexpected considering *E. coli* and *B. subtilis* are gram-negative and gram-positive bacteria, respectively, and have evolved separately for billions of years. In contrast, a sensitive Infernal search using either of the potential *B. subtilis* TPP riboswitches does yield hits to other TPP riboswitches, including the *E. coli thiM* riboswitch (**Fig 5**). These findings highlight FURNA's capability for function annotations of low-homology RNAs using its sensitive sequence search option, providing a unified interface for obtaining functional information on a new RNA of interest.

## Discussion and conclusions

We introduce FURNA, the first comprehensive structure database for ligand–RNA interactions and RNA function annotations. Compared to existing RNA structure and function databases, FURNA stands out in several ways. Firstly, it is the only database to utilize standard function vocabularies (GO terms and EC numbers) for the annotation of RNA tertiary structures. Secondly, it outlines ligand–RNA interactions based on biological assembly, which enhances the investigational context of interactions within the complete RNA-containing complex. Thirdly, FURNA offers user-friendly database search capabilities at varying levels of sensitivity, ensuring its relevance in annotating even remote RNA homologs. Fourthly, its data curation code is modular and fully open source, thereby simplifying regular data updates and future development. These unique aspects of FURNA position it as a valuable resource for the biological community, aiding in summarizing known RNA biological functions, creating functional hypotheses for poorly characterized RNAs, and developing new algorithms for ligand-RNA docking, virtual screening, and structure-based RNA function annotation. Nonetheless, FURNA does present a challenge in its lack of a clear definition of the biological relevance of ligand–RNA interactions (i.e., distinguishing biologically meaningful ligands from crystallization buffer components), an issue we plan to address in our future work.

## Materials and methods

Each entry in FURNA corresponds to 1 RNA chain in the PDB database. To this end, we first download the mmCIF files of all structures containing nucleic acid from the PDB database and split them into individual chains using a modified version of the BeEM tool [33]. An RNA

(A)

| # | Hit | Hit length | Strand | E-value | Homologs to hit |
|---|-----|-----------|--------|---------|-----------------|
| 1 | 2hoo:A Crystal structure of an E. coli thi-box riboswitch bound to benfotiamine; thi-box riboswitch | 83 | + | 1.8e-26 | |
| 2 | 2gdi:X Crystal structure of thiamine pyrophosphate-specific riboswitch in complex with thiamine pyrophosphate; TPP riboswitch | 80 | + | 3.8e-26 | 2gdi:Y, 7tzs:X, 7tzs:Y, 4nya:A, 4nya:B, 7tzr:X, 7tzr:Y |
| 3 | 7tda:A Crystal structure of the E. coli thiM riboswitch in complex with thiamine pyrophosphate, manganese ions; thiM TPP riboswitch RNA (80-MER) | 80 | + | 3.7e-23 | 7tdc:A, 7tdb:A, 4nyd:A, 4nyb:A, 4nyc:A |
| 4 | 2hoj:A Crystal structure of an E. coli thi-box riboswitch bound to thiamine pyrophosphate, manganese ions; thi-box riboswitch | 79 | + | 1.5e-22 | 4nyg:A |
| 5 | 7tzu:A Crystal structure of the E. coli thiM riboswitch bound to 1-(4-(piperazin-1-yl)pyridin-3-yl)-N-(quinoxalin-6-ylmethyl)methanamine (linked compound 38); RNA (79-MER) | 79 | + | 7.1e-22 | 2hol:A, 7tzt:A |
| 6 | 2hom:A Crystal structure of an E. coli thi-box riboswitch bound to thiamine monophosphate; thi-box riboswitch | 80 | + | 9.9e-22 | |
| 7 | 7td7:A Crystal structure of an E. coli thiM riboswitch bound to thiamine, manganese ions; thiM riboswitch RNA (78-MER) | 78 | + | 5.7e-21 | |
| 8 | 3k0j:E Crystal structure of the E. coli ThiM riboswitch in complex with thiamine pyrophosphate and the U1A crystallization module; RNA (87-MER) | 87 | + | 4.1e-20 | 3k0j:F |
| 9 | 2hok:A Crystal structure of an E. coli thi-box riboswitch bound to thiamine pyrophosphate, calcium ions; thi-box riboswitch | 78 | + | 6.9e-20 | |
| 10 | 8f4o:A Apo structure of the TPP riboswitch aptamer domain; TPP riboswitch aptamer domain | 72 | + | 3e-17 | |

(B)

| # | Hit | Hit length | Strand | E-value | Homologs to hit |
|---|-----|-----------|--------|---------|-----------------|
| 1 | 7tda:A Crystal structure of the E. coli thiM riboswitch in complex with thiamine pyrophosphate, manganese ions; thiM TPP riboswitch RNA (80-MER) | 80 | + | 8.2e-09 | 7tdc:A, 7tdb:A, 4nyd:A, 4nyb:A, 4nyc:A |
| 2 | 2hoo:A Crystal structure of an E. coli thi-box riboswitch bound to benfotiamine; thi-box riboswitch | 83 | + | 3.5e-08 | |
| 3 | 2hoj:A Crystal structure of an E. coli thi-box riboswitch bound to thiamine pyrophosphate, manganese ions; thi-box riboswitch | 79 | + | 8.3e-08 | 4nyg:A |
| 4 | 2gdi:X Crystal structure of thiamine pyrophosphate-specific riboswitch in complex with thiamine pyrophosphate; TPP riboswitch | 80 | + | 1.8e-07 | 2gdi:Y, 7tzs:X, 7tzs:Y, 4nya:A, 4nya:B, 7tzr:X, 7tzr:Y |
| 5 | 2hom:A Crystal structure of an E. coli thi-box riboswitch bound to thiamine monophosphate; thi-box riboswitch | 80 | + | 1.9e-07 | |
| 6 | 7tzu:A Crystal structure of the E. coli thiM riboswitch bound to 1-(4-(piperazin-1-yl)pyridin-3-yl)-N-(quinoxalin-6-ylmethyl)methanamine (linked compound 38); RNA (79-MER) | 79 | + | 2e-07 | 2hol:A, 7tzt:A |
| 7 | 2hok:A Crystal structure of an E. coli thi-box riboswitch bound to thiamine pyrophosphate, calcium ions; thi-box riboswitch | 78 | + | 3.7e-07 | |
| 8 | 7td7:A Crystal structure of an E. coli thiM riboswitch bound to thiamine, manganese ions; thiM riboswitch RNA (78-MER) | 78 | + | 5.1e-07 | |
| 9 | 3k0j:E Crystal structure of the E. coli ThiM riboswitch in complex with thiamine pyrophosphate and the U1A crystallization module; RNA (87-MER) | 87 | + | 8.6e-07 | 3k0j:F |
| 10 | 8f4o:B Apo structure of the TPP riboswitch aptamer domain; TPP riboswitch aptamer domain | 65 | + | 5.8e-05 | |

**Fig 5. FURNA database sequence search results for TPP riboswitches from _B. subtilis_.** (A) Top 5 Infernal search hits for the _ykoF_ riboswitch. (B) Top 5 Infernal search hits for the _tenA_ riboswitch.

chain is defined by possessing more ribonucleotides than deoxyribonucleotides and amino acids. RNA chains with 10 or more nucleotides become entries in FURNA, but oligo-ribonucleotide fragments with fewer than 10 nucleotides are only included as "ligands" if they bind to an RNA chain with 10 or more nucleotides. The curation of an RNA chain involves several steps: annotating GO terms and EC numbers, mapping RNA-protein binding motifs, extracting RNA–ligand interactions, and assigning RNA secondary structures.

## Rfam family matching

Rfam families are matched to an RNA chain by 2 approaches. First, we use the Rfam-PDB mapping provided by the Rfam database (https://ftp.ebi.ac.uk/pub/databases/Rfam/CURRENT/Rfam.pdb.gz). Second, if a PDB chain is not included in the mapping file, we search its RNA sequence against the most current version of the Rfam database (Rfam 14.10, with covariance models for 4170 families, https://ftp.ebi.ac.uk/pub/databases/Rfam/CURRENT/Rfam.cm.gz) using Infernal [41]. This Infernal search utilizes the parameters: cmsearch --cpu 4 -Z 549862.597050 --toponly, where the search space size parameter 549862.597050 is the same as that used by the Rfam database. Regardless of which approaches are used, Rfam families for an RNA chain are shown in ascending order of their E-values.

This workflow is not perfect, but reflects the best results that can be attained using automated procedures with currently available databases. As an example of a present false negative, PDB 1vc6 chain B is a full-length experimental structures of hepatitis delta virus (HDV) ribozyme, but it is not matched to the Rfam family for HDV ribozyme (RF00094) by either of the 2 approaches mentioned above, because aligning 1vc6 sequence to the covariance model of RF00094 results in a very high (E-value = 2,200). This is probably because an approximately 20-nucleotide long fragment corresponding to the middle of the RF00094 covariance model is absent in the experimental structure (**S5 Fig**).

## Catalytic site annotation

The nucleotides at the active site of ribozymes are annotated based on the Ribocentre [42] database, which to our knowledge is the only curated database for ribozyme active sites available as of this writing. Among all 21 types of ribozymes collected by Ribocentre, detailed active site information on the RNA 3D structure is available only for a subset of 15 types of ribozymes (**S3 Table**). For each of these 15 types of ribozymes, active site nucleotides are annotated for only 1 representative experimental structure. To extend the Ribocentre database annotation to all members of these 15 ribozyme types, we implemented a template-based approach using US-align [43]. Briefly, the representative experimental structure of a ribozyme type is used as the query by US-align to search through all FURNA RNA chains that share the same Rfam families as the target ribozyme type (**S3 Table**). FURNA RNAs with significant structure similarity (TM-score $\geq$ 0.45 [44]) to the query RNA are collected. Additionally, if the query ribozyme has <100 nucleotides, all FURNA RNAs without Rfam families are also searched by US-align to get similar FURNA RNAs with TM-score $\geq$ 0.45, as the covariance models of the Rfam families for short ribozymes may not match the FURNA RNA sequence with significant E-values as shown in the HDV ribozyme example mentioned in the previous paragraph. Nucleotides on the FURNA RNAs that are aligned to the active sites on the query ribozyme structure are marked as putative active site members.

## GO term and EC number annotation

We employ 2 complementary strategies to obtain GO terms for an RNA chain. First, we transfer the GO terms related to each Rfam family (http://current.geneontology.org/ontology/

external2go/rfam2go) of the query RNA. Second, we map RNA chains to RNAcentral sequences based on the mapping file provided by RNAcentral (http://ftp.ebi.ac.uk/pub/databases/RNAcentral/current_release/id_mapping/database_mappings/pdb.tsv). If the RNA-central entry has GO terms in the Gene Ontology Annotation (GOA) project (http://ftp.ebi.ac.uk/pub/databases/GO/goa/UNIPROT/goa_uniprot_all.gaf.gz), we also transfer these GO terms to the FURNA entry. We utilize Graphviz [35] to plot the direct acyclic graphs showcasing the relationships among an RNA's GO terms (including their parent terms). For the subset of RNAs with annotated catalytic activities, we convert their EC numbers from GO terms using the EC2GO mapping (https://www.ebi.ac.uk/GOA/EC2GO). For RNA-binding proteins, their UniProt accessions, GO terms, and EC numbers are directly obtained through the SIFTS [10] database.

## RNA-protein binding motif mapping

To identify RNA motifs corresponding to known recognition sites for RNA-binding proteins, we download the position weight matrices (PWMs) for all 1,583 protein-binding motifs from the latest ATtRACT database (version 0.99β). These motifs and the query RNAs collected by FURNA are grouped by species. Here, we extract the species information of an RNA chain from the respective mmCIF file, specifically from records such as "gene_src_ncbi_taxonomy_id," "ncbi_taxonomy_id," "pdbx_gene_src_ncbi_taxonomy_id," or "pdbx_ncbi_taxonomy_id." For any species that has at least 1 ATtRACT motif and 1 FURNA RNA chain, we download its transcriptome from the NCBI FTP site (ftp://ftp.ncbi.nlm.nih.gov/genomes/all/annotation_releases/) to determine its background distribution of the 4 nucleotide types (A, C, G, and U). This background information is ascertained using the fasta-get-markov program of the MEME suite [45]. Subsequently, this background file is used by the FIMO program [46] of the MEME suite when it searches the motif PWMs against the FURNA RNAs with the parameters:--norc --bfile, to enable the motifs to align with the RNAs.

## Ligand–RNA interaction extraction

For each query RNA included in the FURNA database, we gather its interaction partners from the mmCIF format biological assembly file (i.e., biounit) that contains the pertinent RNA chain. As an example, the asymmetric unit of PDB 1a9n (the spliceosomal U2B"-U2A' protein complex bound to a fragment of U2 small nuclear RNA) contains 6 chains, which comprises 4 protein chains (Chains A, B, C, and D) and 2 RNA chains (Chains Q and R). This PDB correlates with 2 different biological assemblies: assembly 1 includes chains A, B, and Q; assembly 2 incorporates chains C, D, and R. Consequently, to extract ligand–RNA interactions for 1a9n chain R, we only consider assembly 2.

Starting from the biological assembly file selected for a query RNA, we employ a modified version of the BeEM program [33] to split it into different chains. For each chain, we further split the macromolecule part and the small molecule parts, where the former and latter are labeled by numerical values and a period ("."), respectively, in the "label_seq_id" record of the mmCIF file. Next, we collect all non-water ligands from all chains in the mmCIF file, including small molecules and metal ions, proteins, DNAs, and RNAs (excluding the query RNA). For each query RNA-ligand pair, we calculate all inter-molecular atomic contacts, i.e., atom pairs within the sum of the van der Waals radii plus 0.5 Å, among non-hydrogen atoms. We label a nucleotide on the query RNA as a ligand-binding residue if it has 2 or more inter-molecular atomic contacts with a ligand. We group any collection of 2 or more ligand-binding residues for the same query RNA–ligand pair into a binding site. Ligands without a binding site are excluded.

For a small molecule ligand, we extract the name, synonyms, chemical formula, and linear descriptions (including SMILES, InChI, and InChIKey) from the Chemical Component Dictionary (CCD) provided by the PDB database. We perform mappings from PDB ligand IDs (i.e., CCD IDs) to ligand IDs in the ChEMBL, DrugBank, and ZINC databases using the Uni-Chem database [47]. For protein ligands, we retrieve their GO terms, EC numbers, species, and UniProt accessions from the SIFTS [10] database. For DNA ligands, we retrieve the species from the mmCIF file of the asymmetric unit, similar to how we obtain species information for RNA chains.

### RNA secondary structure assignment

FURNA assigns RNA secondary structures in dot-bracket format for canonical base pairs (Watson–Crick pairs and G:U Wobble pairs) in the experimental 3D structure, using 2 complementary methods: CSSR [48] and DSSR [49]. CSSR is our recently developed method optimized for coarse-grained RNA structures. It can assign secondary structures even when the nucleotides have missing atoms. Conversely, DSSR only functions when the nucleobase of the nucleotide is fully atomic and its RMSD to the standard nucleobase conformation [50] is less than 0.28 Å. Due to these stringent requirements, DSSR-assigned secondary structures might have missing positions compared to the input RNA. To ensure consistency between DSSR input and output, we utilize Arena [51] to fill in missing atoms and rectify unphysical nucleobase conformations for all RNA chains before we execute the DSSR assignment. For an RNA-RNA interaction involving 2 RNA chains, we assign secondary structures to both the individual RNAs and the RNA pair.

### Infernal database construction

For users to perform sensitive Infernal searches of query RNA sequences through FURNA, a database in the Infernal [41] format must be preconstructed. To accomplish this, we first obtain a nonredundant set of RNAs, which is generated by collapsing multiple FURNA RNAs with identical sequences into 1 entry. For each RNA in the nonredundant set, the CSSR-assigned secondary structure in dot bracket is collected. Pseudoknots present in the secondary structures are removed by an incremental length (IL) approach, where nonconflicting paired regions are added one by one, starting with the longest paired region [52]. Subsequently, the secondary structure and sequence are converted by the "cmbuild" tool of the Infernal package into the uncalibrated Infernal format covariance model. This covariance model is then calibrated by the "cmcalibrate" tool of the Infernal package. The calibrated covariance models for all nonredundant FURNA RNA chains are concatenated into the Infernal format database. This database can be utilized by the "cmscan" tool of the Infernal package, allowing a user to perform Infernal searches of query RNA sequences through FURNA.

### US-align database construction

Since conducting a tertiary structure search of all RNA chains in FURNA is more time-consuming than a sequence search, 2 procedures are implemented to reduce the size of the structure database used for US-align search. First, the nonredundant set of RNAs with nonidentical sequences is isolated, from which the coordinates of the C3' atoms are extracted. The exclusion of atoms other than C3' does not affect US-align, which only considers C3' atoms for RNA structure alignment. Second, we utilize the qTMclust tool [43] from the US-align package to cluster the structures of the nonredundant RNAs. This results in a set of representative RNA structures with a pairwise TM-score [44] less than 0.5. These representative RNA structures form the US-align database. When a user carries out an RNA structure query through the

FURNA website, this query structure will be searched using US-align through the database of representative structures to report the top 100 hits with the highest TM-scores. Meanwhile, the RNAs belonging to the same structure clusters will also be reported.

## Supporting information

**S1 Fig. Examples of binding sites for biologically relevant cations versus phasing additives.** Shown are X-ray structures of group I intron P4-P6 domains from *Tetrahymena thermophila* in complex with $Mg^{2+}$ and cobalt hexammine (III). (A) Structure determined with $Mg^{2+}$ (yellow spheres) and Cobalt hexammine (III) (magenta spheres), PDB 1gid chain A. (B) Structure determined with $Mg^{2+}$ (blue spheres) only, PDB 6d8o chain A. (C) Overlap of the 2 structures. Magenta arrows in panels (A) and (B) indicate cobalt hexammine (III) binding sites, which are completely different from $Mg^{2+}$ binding sites.
(TIF)

**S2 Fig. Distributions of F1-scores for the similarities of GO annotations between interacting versus random RNA-RNA pairs or interacting versus random RNA-protein pairs.** Here, the F1-score of GO annotations between molecule *A* and molecule *B* is calculated as:

$$F1 = \frac{2 \cdot |GO_A \cap GO_B|}{|GO_A| + |GO_B|}$$

Here, $GO_A$ and $GO_B$ are the set of GO terms (including parent terms) in molecule *A* and molecule *B*, respectively. The solid horizontal bars inside each violin show the mean F1-score. The values above the violin plots show the *p*-value of Wilcoxon rank sum tests between adjacent violins. We observe that just as for proteins, inter-molecular interactions provide a substantial amount of information regarding BP and CC terms, but not for MF terms (as might be expected, as interacting pairs should co-localize in the cell and be involved in the same pathway but will typically not have the same function at the molecular level). *P*-value <0.001 and 0.01~0.001 are marked by \*\*\* and \*\*, respectively.
(TIF)

**S3 Fig. Web page for protein–RNA interaction between human splicesomal U2B" protein (PDB 1a9n chain B) and U2 small nuclear RNA (PDB 1a9n chain Q, https://seq2fun.dcmb. med.umich.edu/furna/pdb.cgi?pdbid=1a9n&chain=Q&lig3=protein&ligCha=B&ligIdx=0).**
(TIF)

**S4 Fig. Web page for a hammerhead ribozyme chain (PDB 3zp8 chain A, https://seq2fun. dcmb.med.umich.edu/furna/pdb.cgi?pdbid=3zp8&chain=A).**
(TIF)

**S5 Fig. Multiple sequence alignment for the experimental structure of HDV ribozyme (PDB 1vc6 chain B, first row) and representative sequences of the Rfam family RF00094 "Hepatitis delta virus ribozyme" used to build the RF00094 covariance model.** The covariance model region that is absent in the experimental structure is highlighted by a black box.
(TIF)

**S1 Table. Top small molecule compounds in FURNA, excluding monatomic ions.**
(DOCX)

**S2 Table. The TPP riboswitches used for our case study.**
(DOCX)

**S3 Table. Annotations of active site nucleotides for the 15 types of ribozymes currently covered by the Ribocentre database.**
(DOCX)

## Acknowledgments

We thank Dr. Xiaoqiong Wei for insightful discussions.

## Author Contributions

**Conceptualization:** Chengxin Zhang.

**Funding acquisition:** Lydia Freddolino.

**Investigation:** Chengxin Zhang, Lydia Freddolino.

**Methodology:** Chengxin Zhang.

**Software:** Chengxin Zhang.

**Writing – original draft:** Chengxin Zhang, Lydia Freddolino.

**Writing – review & editing:** Chengxin Zhang, Lydia Freddolino.

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
