## [Editor Report · Decision Letter 0]

15 Dec 2023

Dear Dr Freddolino, 

Thank you for submitting your manuscript entitled "FURNA: a database for function annotations of RNA structures" for consideration as a Methods and Resources Article by PLOS Biology. Please accept my sincere apologies for the long delay in getting back to you as we consulted with an academic editor about your submission.

Your manuscript has now been evaluated by the PLOS Biology editorial staff, as well as by an academic editor with relevant expertise, and I am writing to let you know that we would like to send your submission out for external peer review.

Once your full submission is complete, your paper will undergo a series of checks in preparation for peer review. After your manuscript has passed the checks it will be sent out for review. To provide the metadata for your submission, please Login to Editorial Manager (https://www.editorialmanager.com/pbiology) within two working days, i.e. by Dec 17 2023 11:59PM.

Kind regards,

Richard

Richard Hodge, PhD

rhodge@plos.org

PLOS

---

## [Decision Letter · Decision Letter 1]

13 Feb 2024

Dear Dr Freddolino,

Thank you for your patience while your manuscript "FURNA: a database for function annotations of RNA structures" was peer-reviewed at PLOS Biology as a Methods and Resources Article. Please accept my sincere apologies for the long delays that you have experienced during the peer review process. Your manuscript has now been evaluated by the PLOS Biology editors, an Academic Editor with relevant expertise, and by three independent reviewers. 

In light of the reviews, which you will find at the end of this email, we would like to invite you to revise the work to thoroughly address the reviewers' reports.

As you will see, the reviewers are generally positive about the FURNA database and they think it will be a useful resource for the field. The reviewers raise some specific comments to improve the clarity and reporting in the manuscript and to improve the overall utility of the resource, including the ability to download coordinates of the structures and providing a view of the catalytic pocket for catalytic RNAs.

Given the extent of revision needed, we cannot make a decision about publication until we have seen the revised manuscript and your response to the reviewers' comments. Your revised manuscript is likely to be sent for further evaluation by all or a subset of the reviewers.

**IMPORTANT - SUBMITTING YOUR REVISION**

*Re-submission Checklist*

*Published Peer Review*

*PLOS Data Policy*

*Blot and Gel Data Policy*

Sincerely,

Richard

Richard Hodge, PhD

rhodge@plos.org

REVIEWS:

Reviewer #1: The manuscript by Zhang and Freddolino describes the development of a database called "FURNA" (Functions of RNA) that seeks to provide a comprehensive annotation of RNA structures in the PDB database. The functional components of the RNA that the authors focus upon are the RNA's interactions with ligands including small molecules, ions, proteins and other DNAs. In the database, the authors provide an easy-to-use search interface with a number of descriptors that enables a user to readily find a particular class of RNAs or an individual sequence, or a set of RNAs that interact with a specific ligand (probably the most useful). This list of hits then includes a set of links including the PDB, Rfam and a GO database. This functional annotation of RNA structures is unique to RNA databases and thus should be a useful tool for many RNA researchers.

Overall the manuscript is mostly well-written and provides a good walk-through of their database and its features. However, I do have a few suggestions to improve the manuscript and database as detailed below.

1. line 48-49. RNA quarternary structure describes a set of RNA-RNA interactions (akin to proteins; refer to an excellent review by Jones and Ferre-D'Amare on this topic) and not RNA-ligand interactions. Please rephrase.

2. Line 68-70. I would consider osmium hexamine, iridium hexamine and cobalt hexamine (all frequent ligands in RNA structures) to be ionic compounds as they are all metallic ions that have a set of inner sphere coordinations. These amines are the same as waters in Mg(2+) or other transition metals, but much more slowly exchanging, which is why we observe them with high occupancy in the density map.

3. Lines 160-165. I found this whole paragraph to be confusing. I think that I understand the point the authors are conveying (the ability of FURNA to find examples of weakly related RNAs to the search term), but how this idea is conveyed in the paragraph makes it difficult to determine what the authors specifically did in their search. Further, would these weaker hits be present in Rfam in their list of RNA structures? (As an RNA scientist, I would never use BLAST to find other representatives of the RNA).

In looking at the database, I found a few things I think could be improved.

1. In a search on a riboswitch that binds a ligand, the page that yields different structures is very helpful and is well-annotated. I particularly like the third view that yield details of the RNA that directly interact with the ligand. However, when I tried to download the coordinates of this structure, all I got was the ligand (not useful). Being able to download these coordinates I think is crucial, because viewing in the database viewer is very limited as compared to Chimera or PyMol. Considering revising this feature.

2. Searching on the hammerhead ribozyme (5EAQ) yields a very confusing description of the structure. Specifically, the authors look at the RNA from the perspective of the enzyme strand and consider the substate strand a ligand. This leads to some strange views of the RNA structure as just the enzyme strand (top view), which really does not reflect the biological assembly. Further, the most important aspect of these RNAs is the catalytic pocket and the nucleotides surrounding the cleavage site—a view not given. Thus, this makes this database really not very useful for catalytic RNAs and understanding the function of their active sites. This issue is rife throughout the database for catalytic RNAs.

Reviewer #2: The MS "FURNA: a database for function annotations of RNA structures" describes a newly built database that is filling up so far empty niche of functional annotations of experimentally determined RNA structures. I actually enjoyed reading the MS, it is well written and technically sound, the authors correctly overview the existing structural databases.

I have no serious comments to its publication, just a few notes. 

I appreciate that the authors went into trouble with generating symmetry-built biological units. 

The statement "DSSR only functions when the nucleobase of the nucleotide is fully atomic and its RMSD to the standard nucleobase conformation is less than 0.28 Å." on pg. 9 is unclear. What is "standard nucleobase conformation"? In general, I would warn agains using DSSR structural classifications, e.g., its base par assignments are very unreliable, base morphology often sheer nonsense. 

I would suggest to use Mol* instead of jsmol as it speeds up rendering of large structures but I do understand difficulties with poorly documented Mol* software. 

The following suggestion is much stronger: use mmCIF instead of PDB format as the the latter is obsolete while the former opens up opportunities to add your own annotations to individual flat files in form of custom-built mmCIF categories. 

Minor thing, on page 2 should be "… many proteins, they ARE critical for the folding of many …"

Reviewer #3: FURNA, by Chengxin Zhang and Lydia Freddolino, collects together a large number of data items concerning RNA chains in the Protein Data Bank. The database is updated every week. The data items are helpful to have pre-computed and to have them available in a central place. Basic search of the annotations is available. More sophisticated searches based on sequence and 3D structure are also provided. 

Comments

Line 20, 16,000 RNA chains, not 1,600.

Line 63, they are critical

Figure 1, it would be great if the text was characters that can be copied and not an image that pixelates when you zoom in, or that has visual artifacts that make some lines harder to read. 

Figure 1, can the website be used to generate these lists, counts, and histograms? If not, it's OK, just be clear.

Figure 2 is impossible to read on a printed page, and in the PDF it pixelates and is barely legible. In Panel A, it would be sufficient to show one entry from the summary table, at a resolution where it can be read and explained. Panel B could be split into two images, or some section left out.

FURNA site, it would be nice if dataset.txt had exactly one line per PDB entry, instead of the name running onto the next line. It seems that this is the same breakpoint as in the hover text. It might also be nice if dataset.txt had a more self-documenting name, like FURNA_dataset.txt.

FURNA site, it's a bit strange that the link to the FURNA page about a chain is in the line number. That won't be obvious to people at first, that they should even bother clicking it.

FURNA site, the mapping of PDB chains to Rfam families is not well enough calibrated, because individual chains often map to multiple Rfam families. For example, bacterial SSU chains map to bacterial, eukaryotic, and archaeal SSU Rfam families, plus many other Rfam families of different molecules. One example is 6h58 chain aa, but there are many others. That's not as helpful as getting the domain right and the molecule right. In dataset.txt, the Rfam hits are listed in lexicographic order, not apparently in order from best hit to worst, so the user can't guess what is the best hit. In some cases, the Rfam site for the family does not map to any chains in PDB, but FURNA maps those chains to the Rfam family. A more stringent cutoff could be used. Or, once some good hits are found, all worse hits could be dis-regarded.

Figure 3 is very hard to read on a printed page. Reducing the number of lines of output, splitting the figure, could help.

Line 151, with various types of searches going on, and many possible starting points for the searches, it would be good to clarify here, and not just in Table S2, that this is a sequence-based search, and that a genomic sequence of length 200 was input (as opposed to the sequence of some RNA chain from PDB). It would also be good to make clear that FURNA does not allow a user to search an entire genome, but rather the user must locate a short sequence of interest from a genome and then search it against Infernal models derived from PDB chains. As far as I can tell, FURNA is not a tool for annotating full genomes with RNAs whose 3D structures are in PDB.

FURNA site, in the 3D structure search results, it would be nice if hovering over a match would show the title of the structure.

Line 199, please be clear about the criteria for a hit on an Rfam family, because as mentioned above, hits are reported that don't make much sense. 

Line 268, please be clear how pseudoknots are removed, because there are different strategies, described in an article by Rob Knight and Sandra Smit, that can lead to different Watson-Crick pairs being removed.

---

## [Decision Letter · Decision Letter 2]

13 Jun 2024

Dear Dr Freddolino,

Thank you for your patience while we considered your revised manuscript "FURNA: a database for function annotations of RNA structures" for publication as a Methods and Resources at PLOS Biology. This revised version of your manuscript has been evaluated by the PLOS Biology editors, the Academic Editor and a subset of the original reviewers.

Based on the reviews and our Academic Editor's assessment of your revision, we are likely to accept this manuscript for publication, provided you satisfactorily address the following data and other policy-related requests.

IMPORTANT - please attend to the following:

a) Please change your title to: "FURNA: a database of functional annotations of RNA structures"

b) Per journal policy, any code or database generated during the course of an investigation needs to be made available without any restrictions upon publication. Please ensure that the code is sufficiently well documented and reusable, and that your Data Statement in the Editorial Manager submission system accurately describes where your code can be found. As the database that you have generated is of vital importance to your manuscript, its deposition is required for acceptance. PLEASE NOTE that we cannot accept an institute URL as the sole location of your database, as this could be changed after publication. It would be preferable if you can host the entire database elsewhere. You also need to archive a version of your database to Zenodo. Once you do this, it will generate a DOI number, which you will need to provide in the Data Accessibility Statement. See the process for doing this here: https://docs.github.com/en/repositories/archiving-a-github-repository/referencing-and-citing-content

We expect to receive your revised manuscript within two weeks. 

*Published Peer Review History*

*Press*

Sincerely,

Suzanne

Suzanne De Bruijn, PhD, 

Associate Editor

sbruijn@plos.org

PLOS Biology

Reviewer remarks:

*Reviewer #2:

[identifies himself as Bohdan Schneider]

 I read carefully author's answers to questions and suggestions by all three referees and appreciate detailed answers and explanations provided by the authors. My only remaining suggestion would be to to consider full implementation of mmCIF instead of relying on the obsolete PDB format. Currently very incomplete and often incorrect annotation of NA features in the public PDB archive needs and hopefully will be supplemented by new mmCIF categories. Considering the fact that no database is ever finished and complete, I am happy to recommend the manuscript for publication. 

*Reviewer #3: 

I appreciate the attention given to the concerns of the reviewers. The manuscript and the database are much improved.

---

## [Editor Report · Decision Letter 3]

24 Jun 2024

Dear Dr Freddolino,

Thank you for the submission of your revised Methods and Resources "FURNA: a database of functional annotations of RNA structures" for publication in PLOS Biology. On behalf of my colleagues and the Academic Editor, Yunsun Nam, I am pleased to say that we can in principle accept your manuscript for publication, provided you address any remaining formatting and reporting issues. These will be detailed in an email you should receive within 2-3 business days from our colleagues in the journal operations team; no action is required from you until then. Please note that we will not be able to formally accept your manuscript and schedule it for publication until you have completed any requested changes.

PRESS

Sincerely, 

Suzanne

Suzanne De Bruijn, PhD, 

Associate Editor

PLOS Biology

sbruijn@plos.org